# Production and Characterization of Non-Isocyanate Polyurethane/SiO_2_ Films Through a Sol-Gel Process for Thermal Insulation Applications

**DOI:** 10.3390/polym11101596

**Published:** 2019-09-29

**Authors:** Natalia E. Noriega, Amanda Carrillo, Santos J. Castillo, María L. Mota

**Affiliations:** 1Institute of Engineering and Technology, Autonomous University of Ciudad Juarez, Av. del Charro 610, Ciudad Juárez, CHIH 32310, Mexico; al164312@alumnos.uacj.mx; 2Department of Research in Physics, University of Sonora, Blvd. Luis Encinas y Rosales S/N, Hermosillo, Sonora 83000, Mexico; santos.castillo@unison.mx; 3CONACYT-Institute of Engineering and Technology, Autonomous University of Ciudad Juarez, Av. del Charro 610, Ciudad Juárez, CHIH 32310, Mexico

**Keywords:** isocyanate-free polyurethane, sol-gel process, silica nanoparticles

## Abstract

The reaction of cyclic carbonates with amines is the most attractive among the synthesis methods for isocyanate-free polyurethane. Non-isocyanate polyurethane films with SiO_2_ NPs fabricated by a sol-gel process are reported, where cyclic carbonates (CC) were produced under mild conditions by CO_2_ insertion in an epoxide complex in the presence of LiCl. A reaction of CC and polyamines was carried out in a low concentration polymer matrix of PVA. The materials were characterized by ^1^H-NMR, FTIR, UV-Vis, SEM, TGA, DTG, and a KD_2_ pro technique. polymer FTIR results are consistent with the literature, even with the use of a non-conventional methodology, where the found chemical interactions values were 3330, 2930 and 1637 cm^−1^. There are differences in the polymers’ morphologies due to the presence and absence of SiO_2_ NPs according to SEM, where the spherical morphology and homogenous particle size distribution of NPs around 100 nm. According to TGA results, all polymers showed their last stage decomposition after 300 °C and polymers with higher concentration of NPs showed even better stability. Due to the obtained results, the polymers have the potential to be used for thermal insulation without negative effect on the environment.

## 1. Introduction

Polyurethane (PU) is one of the most used thermal insulation materials with the greatest potential, in terms of thermal conductivity and diversity of applications. Thermal insulation is in high demand for economic reasons and for energy conservation, being able to provide comfort under sustainable policies. In 1937, Bayer and coworkers [1] discovered the synthesis of PU by reacting a polyester diol and a diisocyanate. As is known, isocyanates are highly toxic and harmful to the environment [2]. Therefore, international standards of quality and environmental safety already ban the use of isocyanates [3]. The authors employed the sol gel method to produce NIPU, a process the directly impacts the reaction time, energy consumption and costs mainly due to the fact it isn’t necessary to use reactors, long process times or vacuum or higher temperatures [4]. Although the most feasible synthesis for NIPU is the reaction between cyclic carbonates (cc) and amines, the current methods require high pressures and temperatures as complex catalysts are used. Besides, this route has low reactivity at room temperature and low yields are obtained [5], while another limitation in the foaming process for PU foams, in which a physical or chemical expansion agent is required.

Cyclic carbonates are versatile regents that in a green approach, can be manufactured from biodegradable oxiranes of low cost and toxicity, but the increasing demand for production due to their lack of commercial availability makes the products more expensive [6]. Meanwhile, polyvinyl alcohol (PVA) is a non-toxic water-soluble biocompatible polymer with wide applications like the synthesis of bioactive materials. Due to the extension of the hydroxyl groups provided by the PVA matrix, NIPUs’ properties can be improved, e.g., the chemical resistance is 1.5–2 times superior to materials with a similar structure without affecting the use of non-renewable resources that put shrimp species at risk, i.e., the use of chitin as an alternative diol extract [1]. With the reinforcement of silica nanoparticles especially in these types of polymer matrices, their thermal, mechanical or other properties can be significantly improved [7]. Nanoparticles with desirable characteristics and low production costs can be manufactured by the sol-gel process. The production of polyurethanes without isocyanates and phosgenes, -named poly(hydroxy)urethanes (PHUs)-, by this more economically feasible methodology, is then more suitable for the preservation of the environment with the development of less toxic products.

Therefore, a new NIPU material was synthesized in high yield from 5-membered cyclic carbonates (5cc) under mild conditions is proposed, with desirable thermal properties, low production costs and low energy consumption by the use of a polymer matrix such as the PVA in aqueous solution at low concentration as an alternative to non-sustainable precursors. The final products offer sustainable production and scalability at the industrial level by being reinforced with silica nanoparticles that can improve the thermal, and mechanical properties of commercial polyurethanes. The nanoparticles are synthesized using the same process. The sol-gel the process directly impacts positively the time and costs mainly due to the fact it isn’t necessary to use complex reactors, vacuum or higher temperatures, which is more suitable for the preservation of the environment in thermal insulation applications.

## 2. Materials and Methods 

### 2.1. General Information 

All solvents and reagents were purchased from commercial sources and used as received for preparing the cyclic carbonates, the SiO_2_ nanoparticles, and the polymers. 1,2-Epoxybutane (C_4_H_8_O), aniline (C_6_H_7_N) and dimethylformamide were supplied by Sigma Aldrich (Toluca, México), with a purity of 99%, ≥99.5 and 99.8%, respectively. Poly(vinyl alcohol) ((CH_2_CHOH)_n_ < 99%) and 1,6-hexamethylenediamine (C_6_H_16_N_2_, 98%) were purchased from Aldrich and lithium chloride (LiCl, <99.69%) from CTR Scientific (Monterrey, México). For silica nanoparticles, tetraethyl orthosilicate (C_8_H_20_O_4_Si) with a purity of 98% was provided by Sigma Aldrich, ammonium hydroxide (NH_4_OH, 28.7%) by Fermont (Monterrey, México) and ethyl alcohol (CH_3_CH_2_OH, 96%) supplied by Golden Bell (Jalisco, México). All reactants were used without further purification. The experiments were carried out in a 100 mL beaker as a closed system, according to the times and set temperatures. Nuclear magnetic resonance spectroscopy (^1^H-NMR), Fourier transform infrared spectroscopy (FTIR), ultraviolet-visible absorption spectroscopy (UV-Vis) and scanning electron microscopy (SEM), besides thermogravimetric analysis (TGA), derivative thermogravimetric analysis (DTG) and a transient line heat source method, were the characterization techniques applied in this research.

### 2.2. Synthesis of 5-Membered Cyclic Carbonates

The first focus of this research was the synthesis of 5-membered cyclic carbonates (5cc) by a new methodology. PVA was used as a matrix, and also as a strong protective group due to its high hydroxyl group content so as not to compromise the epoxide groups and break the appropriate carbon backbones without reducing them to OH groups, instead of chitin as proposed before [8] According to established times and temperatures, 3.6 mL of a 0.125% PVA solution and 12.9 g (15 mol) of 1,2-epoxybutane were dissolved in 20 mL of DMF and placed in a 100 mL beaker to be stirred at 300 rpm for 5 min at a set temperature of 35 °C. The solution remained under constant stirring for an additional twenty minutes at a temperature of 95 °C in the presence of LiCl as a catalyst and this step ended after another 30 min. Then, a solution produced from 0.25 g of aniline and continuous addition of CO_2_ is added to the reaction mixture, allowing the whole synthesis to be carried out after an additional 20 min of stirring and at a temperature of 90 °C. All the amounts of the reagents were chosen according to stoichiometric calculations. The cyclic carbonates resulted produced biphasic mixtures containing a liquid and a precipitated phase, which have the same composition and chemical structure, according to the optical and chemical characterization. At the end of the reactions, the 5cc present as transparent biphasic solutions, but as time passes, the material condenses and only one phase remains. For this work, the cyclic carbonates of the first reaction are denominated as *5cc_l* for the liquid phase, *5cc_p* for the precipitated phase and *5cc* for both phases in one. The cyclic carbonates were not purified and were used as they produced. Six cyclic carbonate replicates were also synthesized following this same procedure. 

The ^1^H-NMR data of the 5cc (300 MHz, CDCl_3_ and CD_3_OD) shows the following peaks: *δ* = 7.96 ppm (s, J = 3981.28 Hz, proton neighboring carbonyl group), *δ* = 7.714 ppm (s, J = 3860.33 Hz, CDCl_3_), *δ* = 4.761 ppm (t, J = 2381.72 Hz, cyclic carbonate proton), *δ* = 4.422 ppm (s, J = 2211.72 Hz, enol group product of incomplete dehydration and carbonate proton attached to it), *δ* = 3.310 ppm (q, J = 1656.58 Hz, CD_3_OD), *δ* = 3.129 ppm (s, J = 1565.38 Hz, methyl group of residual epoxide), *δ* = 2.993 ppm (s, J = 1497.20 Hz, terminal methyl of radical chain that forms 5cc). Figure 1 shows a magnification of the region between 4.9–3.0 ppm for the 5cc.

### 2.3. Synthesis of Silica Nanoparticles

The silica nanoparticles were fabricated by a sol-gel process based on the Stöber method and using a similar procedure as previously reported by [9,10]. The preparation of silica nanoparticles requires 1 mL of distilled water, 1 mL of TEOS, 1 mL of NH_4_OH solution (29%) and 27 mL of ethanol (EtOH), all used with no further purification. First, TEOS and water are placed in a vortexer for 15 s before the addition of ethanol and NH_4_OH solution. The reaction is placed under sonication for 5 min and then added to a 50 mL beaker. After carrying out the reaction on the hotplate at 300 rpm and for 24 h at room temperature, the synthesis ends with the formation of the silica NPs. This same procedure was repeated to obtain four equal samples of silica nanoparticles (see Figure 2). For the posterior chemical and optical analyzes of these samples, the nanoparticles were centrifuged at 15,000 rpm for 5 min under 25 °C and for better separation of the small sizes NPs, the procedure was repeated three times. Afterward, the larger nanoparticles existing in the precipitated part were eliminated and the smallest ones of the supernatant phase kept as desired for the SEM analyzes. For the IR spectroscopy, the pure nanoparticles were dispersed in 3.5 mL of ethanol to avoid their agglomeration from affecting the measurements. The final solutions of NPs were homogeneous, transparent and with a volume of 30 mL, using ethanol as solvent. The NPs were stable in all replicate syntheses and during the reactions. By precipitation, it was possible to separate nanoparticle sizes, where the smallest are the ones of greatest interest.

### 2.4. Synthesis of Non-Isocyanate Polyurethane

The PHU material was polymerized by cycloaddition reaction in the presence of metal catalyst under highly controlled parameters [11,12,13]. Briefly, a 100 mL beaker with 10 mL of DMF was used as the dissolution medium for hexamethylene diamine, 3.8 mL of cyclic carbonates (1 M), 5 mL of LiCl (1 M) and 1.59 g of 1,6-hexamethylenediamine (HMDA) (4.3 mm) at 95 °C at 260 rpm for one hour. The reaction synthesis turns into a single phase, transparent and homogenous. The polymer was manufactured and characterized and is named as P1 in this work. Prior to polymerization, the cyclic carbonates were not purified and were used as they were produced.

The product have the following ^1^H-NMR spectrum (300 MHz, CD_3_COCD_3_): *δ* = 3.186 ppm (m, J = 1274.68 Hz, protons bonded to a hydroxyl radical and to the oxygen of the main chain), *δ* = 2.914 ppm (m, J = 1166.45 Hz, *α* position of amino group), *δ* = 2.516 ppm (t, J = 1006.84 Hz, CH_2_ protons of carbamate group), *δ* = 2.204 ppm (m, J = 882.21 Hz, CH_2_ proton), *δ* = 2.04 ppm (m, J = 819.90 Hz, CD_3_COCD_3_), *δ* = 1.356 ppm (m, J = 542.23 Hz, methyl groups that extend the length of the polymer chain). These same data were obtained for the two isomers obtained in the polymer structure (see isomer *a* in Figure 3).

The whole polymer structure gave the following ^1^H-NMR spectrum (300 MHz, CD_3_OD): *δ* = 4.886 ppm (s, J = 1955.36 Hz, CD_3_OD), *δ* = 3.242 ppm (m, J = 1297.01 Hz, protons bonded to a hydroxyl radical and to the oxygen of the main chain), *δ* = 2.948 ppm (m, J = 1179.94 Hz, α position of amino group), *δ* = 2.565 ppm (s, J = 1026.94 Hz, CH_2_ protons of carbamate group), *δ* = 1.467 ppm (m, J = 586.49 Hz, methyl groups that extend the length of the polymer chain) (see Figure 4).

### 2.5. Synthesis of PHUs/SiO_2_ Materials

To produce the polymeric materials, the silica nanoparticles solutions (supernatant phase) were added to finished reaction mixtures of PHU materials, and therefore, we could analyze the influence of NPs at various concentrations on the properties of the polymers. The polymers with silica nanoparticles are identified as P2 (20%), P3 (30%) and P4 (40%), according to the nanoparticle concentrations with respect to the volume of polymeric material. Three polymers containing silica nanoparticles were manufactured and characterized. The solutions of PHUs changed from being transparent and homogeneous to white and turbid mixtures by the addition of the silica nanoparticles.

### 2.6. Names of Materials 

The names used for the different materials are listed in Table 1. 

### 2.7. Preparation of Samples for Analyzes

For the IR tests, all samples were analyzed as produced, and for the UV-Vis analyzes, the samples were prepared according to the material, in a solution for the cyclic carbonates, on substrates for all polymers and in ethanol dispersion for the silica nanoparticles. SEM results were obtained for all materials in powder form. The SEM analyses were carried out by taking material from the dry samples in substrates with a carbon strip. The TGA/DTG studies were conducted in cyclic carbonates and NIPU material in powder form, while the PHUs/SiO_2_ were studied as they were obtained at the end of the synthesis.

### 2.8. Characterization Techniques

#### 2.8.1. Nuclear Magnetic Resonance (NMR) Spectroscopy

^1^H-NMR spectroscopy was used to determine the chemical structure of the molecules using an Ultrashield plus 500 MHz spectrometer (Bruker, Silberstreifen 4, Germany). In the NMR experiments of the 5cc and PHUs samples, 10 mg of the product were prepared in 0.5 mL of deuterated chloroform (CDCl_3_) and methanol (CD_3_OD) as solvents system to induce the solubility of cyclic carbonates, that were insoluble in these solvents individually. While the PHUs were prepared as the 5cc, but in deuterated acetone (CD_3_COCD_3_) for the two PHU isomers, and methanol for the whole PHU compound.

#### 2.8.2. Fourier Transform Infrared (IR) Spectroscopy

Attenuated total reflectance Fourier Transform infrared (ATR-FTIR) spectroscopy was used to determine the chemical composition of the different materials based on their chemical bonds and functional groups and analyze their structure in the different regions ranging from 400 to 4000 cm^−1^. Measurements were performed on a Nicolet iS10 Infrared Spectrophotometer (Thermo Scientific, Waltham, MA, USA) equipped with a MIRacle attenuated total reflection accessory and 32 scans. The ATR technique in which the samples are directly analyzed and then recovered to avoid losses was used.

#### 2.8.3. Ultraviolet-Visible Microscopy (UV-Vis) Spectroscopy

UV-Vis data was executed in a 6850 Double Beam Ultraviolet-visible spectrophotometer (Jenway, Staffordshire, UK). The optical analyzes of cyclic carbonates, PHUs, silica nanoparticles, and PHUs/SiO_2_ materials were measured at 0.50 very slow precision and 0.2 nm of step scan, in the range of 200 to 1100 nm.

#### 2.8.4. Scanning Electron Microscopy (SEM)

The morphology images of the PHUs, SiO_2_ nanoparticles and PHUs/SiO_2_ materials were taken on a Schottky SU5000 Field Emission (FE-SEM) instrument (Hitachi, Santa Clara, CA, USA) using secondary and backscattered electron detector at an acceleration voltage of 15–20.0 kV under low/high vacuum mode and adjusting different magnifications.

#### 2.8.5. Transmission Electron Microscopy (TEM)

Silica nanoparticles morphologies were also imaged using a 2010F transmission electron microscope (JEOL, Peabody, MA, USA) at an acceleration voltage of 200 kV, with the observations carried out in bright field mode.

#### 2.8.6. Energy Dispersive Spectroscopy (EDS)

EDS microanalysis was obtained by the Energy Dispersive X-ray detector of the SU500 instrument to study the chemical composition of the cyclic carbonates, PHUs, silica nanoparticles, and PHUs/SiO_2_ materials.

#### 2.8.7. Thermogravimetric Analysis (TGA) and Derivate Thermogravimetric Analysis (DTG)

Thermogravimetric analysis was carried out with a TGA/DTG Q600 SDT thermal analyzer (TA instruments, New Castle, DE, USA), under an inert nitrogen atmosphere and 10 °C/min heating rates, in the range of 26–600 °C, to examine the thermal stability, in vacuum-dried samples, of the 5cc, pure polymer, and PHUs/SiO_2_ materials.

#### 2.8.8. Thermal Conductivity (TC) Measurement

Thermal conductivity of silica nanoparticles was determined using a thermal properties analyzer (KD2 Pro-ISO 2008 standards, METER Group, Inc., Pullman, WA, USA), based on the transient line heat source method. A small single needle KS-1 (of 1.3 mm diameter × 6 cm long) was placed inside the material in a controlled atmosphere, which heats the samples for a certain time by applying electrical power to the sensor. Samples tests were performed in a water bath at room temperature for 15 min of measurement and 20 min of stability between tests, to obtain values within the margin of error.

## 3. Results and Discussion

### 3.1. Characterizations of Cyclic Carbonates

#### 3.1.1. FTIR and UV-Vis Spectroscopies

The IR spectra for both phases of 5-membered cyclic carbonates are presented in Figure 5a. The signals show minor intensity variations due to different phases and arrangements, having a similar chemical composition as presented in the literature (Table 2). The spectrum in both phases (5cc_l and 5cc_p), shows an almost total disappearance of the peaks at 905 [14] and 831 cm^−1^ [15,16] accredited to epoxy groups, and new absorption peaks at 3420 cm^−1^ (O–H groups) and 1100 cm^−1^ (carbonate) [17]. The more intense and important band is located in the region around 1700 cm^−1^, assigned to the C=O double bond by stretching, which can confirm the presence of a cyclic carbonate [18]. It presented small displacements associated to a CH_2_ variation in the epoxide used and to the steric effect represented by the PVA matrix by the amount of OH groups present in the medium that is strongly attracted by the carbonyl group.

Besides, the presence of two pairs of bands that interact around 1300 cm^−1^ is noted, due to C–H bonds between the *p*-conjugation system [19] that makes the molecule highly stable. These bands, that are the basis of the chain, show a high definition that confirms the good purity of the material, as is observed in ^1^H-NMR (Figure 1), and judging by the band intensity in the fingerprint and functional group regions, it can be concluded that the product has been adequately isolated without interferences. Regarding the functional group identification of the molecule, we can identify the presence of hydroxyl groups represented as a broad band at 3420 cm^−1^ [19], attributed to the four hydroxyl groups of the molecule. At the same time, the bands around 2900 and at 1390 cm^−1^, are assigned to the C–H bond stretching and bending interactions, respectively. In addition, the intensity of the two bands, both O–H and C–N, show symmetric intensity, being only four groups of each per molecule formed, while the marked intensity of the interaction bands of the *p*-conjugation represents a greater amount of the type of link that extends throughout the structure.

In the UV-Vis spectrum of both phases of 5cc (Figure 5b), the bands show maximum absorptions around 300 nanometers, that can be attributed to the carbon-oxygen double bond (C=O) of the carbonate, similar to propylene carbonate according to [20]. Both liquid and precipitated phases have their maximum absorption band around 300 nm in the wavelength, showing hyperchromicity with little variation depending on the reaction parameters. Due to the fact the chemical and optical studies of both phases of the 5cc is very similar, the subsequent characterizations were made not separately, but using the two phases together, and also for the polymerization of these with amines.

#### 3.1.2. Thermogravimetric Analysis and Derivate Thermogravimetric Analysis

Thermal study of 5cc was performed using the TGA and DTG techniques, with the results shown in Figure 6. The data indicated that were thermally stable up to 700 °C. A multistage decomposition process is observed in Figure 6a, where an initial degradation occurs between 83 and 123 °C (at a weight loss of 7.15% due to humidity and isolated epoxy groups) and is associated with an endothermic peak at 99.54 °C presented in Figure 6b. 

The next degradation step takes place in the range of 123–173 °C at a weight loss of 20.37%, possibly due to the loss of solvent and corresponding to an endothermic peak at 149.19 °C. A third degradation step has a maximum weight loss of 2.69% in the range of 173–700 °C, assigned to the decomposition of the cyclic carbonates and is associated with endothermic peaks at 482.15 and 521.20 °C. This process shows a good purity of the material—as observed in the ^1^H-NMR analysis— with stable behavior that is given by the reaction kinetics of the material.

### 3.2. Characterizations of SiO_2_ Nanoparticles

#### 3.2.1. FTIR and UV-Vis Spectroscopies

Infrared analysis was carried out to confirm the chemical composition of the silica nanoparticles, using a 3.5 mL ethanol dispersion to make the measurements. As shown in Figure 7a, the band around 3400 cm^−1^ is characteristic of the O–H groups existing in ethanol [21], followed by a peak at 1050 cm^−1^, that is assigned to an asymmetric stretching vibration of the Si–O–Si bond [22,23] and another peak for symmetric vibration of Si–O bond at 800 cm^−1^ [24], very close to previous data reported in the literature [25]. On the other hand, the UV-Vis absorption spectrum of the silica nanoparticles (Figure 7b) absorbed in the range of 210–325 nm, with a specific peak for the Si–O–Si bond, which confirms the presence of silica nanoparticles, according to [26].

#### 3.2.2. Scanning Electron Microscopy

Through scanning electron microscopy (SEM), the shapes, distribution and diameter sizes of silicon dioxide nanoparticles (SiO_2_) can be studied. As observed in Figure 8a–c, the NPs have a defined spherical morphology, porous particles and very different sizes due to their stratification in the separation process. Before performing the centrifugation process, the NPs have different diameter sizes, homogeneous distribution and have no agglomeration as we see clearly in Figure 8a. Once centrifuged, the larger sized particles precipitate and the average diameter increases to almost 170 nm (Figure 8b), while the smaller NPs remain mostly in the supernatant part of the solution. Only the nanoparticles of the supernatant phase have a normal distribution with an average size of less than 78 nm and a very low standard deviation (Figure 8c). The smaller the NP, the lower its thermal conductivity (Figure 11b). Due to their homogeneous distribution, small surface areas and narrow and small particle sizes distribution, these monodisperse nanoparticles can serve as the basis for the sustainable polymeric materials of tomorrow.

#### 3.2.3. Electron Dispersive Spectroscopy Analysis

The chemical composition of the manufactured silica nanoparticles (before the centrifugation process) was obtained by elemental analysis and it can be confirmed that they have in their structure the silicon and oxygen elements only (Figure 9), so the nanoparticles are free of impurities. The microanalysis has shown that SiO_2_ nanoparticles do not have another element in their structure (Table 3).

#### 3.2.4. Transmission Electron Microscopy and EDS analysis

The diameter sizes, distribution, shapes, and elements quantification of SiO_2_ nanoparticles after centrifugation (supernatant phase) could also be obtained by TEM spectroscopy. Spherical nanoparticles with particle size distribution between 60–80 nm were observed in the images (Figure 10a,b). These results and their homogenous distribution are in good agreement with SEM observations of the supernatant phase, and it is possible to confirm the presence of only silica nanoparticles according to the elemental analysis (Table 4), which were analyzed in a diluted solution. The support grids for analysis are made of copper and carbon, so these elements are not considered.

#### 3.2.5. Thermal Conductivity

Using the thermal analysis equipment and the transient line heat source method, the nanoparticles of silicon dioxide were analyzed before the separation process by centrifugation and after centrifugation (in the supernatant phase), to know the variations in their thermal conductivity. In the non-centrifuged nanoparticles (Figure 11a), an average thermal conductivity of 0.31 W/mK was obtained, according to a sampling of 34 datapoints, with a non-normal distribution, while for the nanoparticles after centrifugation (supernatant phase), the thermal conductivity was reduced to 0.203 W/mK, with 22 sampling datapoints and a normal distribution (Figure 11b). 

Due to the fact the nanoparticles in the supernatant phase have a smaller size and a smaller contact area between them, the thermal conductivity is lower and, therefore, the insulating capacity is better [23,27,28].

### 3.3. Characterizations of Non-Isocyanate Polyurethane Materials

#### 3.3.1. FTIR and UV-Vis Spectroscopies

The IR and UV-Vis spectroscopy results are presented below in order to analyze the polyhydroxyurethanes (see Figure 12a,b). The cyclic carbonates are opened by secondary amines and the signals arising from carbonates changed from 1700 cm^−1^ to 1637 cm^−1^ due to a urethane bond [29] forming hydroxyurethanes with cyclic carbonates (Figure 12a). The peak at 1637 cm^−1^ is assigned to the stretching vibrations of the carbonyl groups of urethane bonds, indicating the presence of a hydroxyurethane linkage, as identified by [30] Another band at 3330 cm^−1^ can be attributed to the hydroxyl groups [11] of the polymeric matrix that occurs by the ring-opening polymerization -offered by PVA-, and to N–H stretching [31,32] very close to the reference value (see Table 5). Intensity changes as a function of time are noted, which prove the polymerization process has occurred. The N–H stretching band at 3330 cm^−1^ is not as wide as reported in the literature, so it is suggested that the urethane bonds form few hydrogen bonds, due to the fact PVA forms more of these links with the chains of hydroxyurethanes.

Other bands were obtained in 2930 cm^−1^ and 2855 cm^−1^, attributed to moderately strong bond stretching between carbon-hydrogen, while others at 1585, 1470, 1398 and 1360 cm^−1^, are originated by a flexion interaction between the same type of carbon-hydrogen bond [33].

In the UV-Vis spectrum of P1, the material has maximum absorption at 300 nm, attributed to the electronic transitions of the urethane bonds, as observed in Figure 12b. In addition, the carbon-oxygen double bond (C=O) of the urethane bond [34], which is the chromophore that absorbs the energy, joins the NH_2_ amine, forming a saturated auxotrophic group that influences the wavelength and the intensity of the maximum of absorption. Such a carbon-oxygen double bond is the most predominant of the urethane base chain, in its π or base state. Using the Tauc theory for the bandgap estimation [35] or P1, a value of 3.93 eV was found (as can be seen in Figure 13), which is within the range of 2.90–3.94 eV established for polyurethanes band [34]. In a similar way, this same Tauc method procedure was carried out for the liquid and precipitated phases of the cyclic carbonates where values of 3.89 eV and 3.96 eV were obtained, respectively.

#### 3.3.2. Scanning Electron Microscopy

The surface morphology of P1 material was evaluated by Scanning Electron Microscope and as can be appreciated in Figure 14a, it is a high porosity and small pore material, with physical properties that allow the material to be foamed. Homogeneity in the particle sizes can be observed, as well as a lack of transparency and no agglomeration between them, as observed in Figure 14b.

#### 3.3.3. Thermogravimetric Analysis and Derivate Thermogravimetric Analysis

TGA and DTG techniques were executed to know the thermal behavior of P1, which represents he obtained polyhydroxyurethane (see Figure 15). P1 exhibited thermal stability up to 600 °C. According to the thermo-gravimetric analysis of P1, there is a multistage decomposition process with similar behavior to the literature report [31], given by intermediates that did not react, such as amines and carbonates (Figure 15a). A first degradation occurs before 100 °C due to humidity release with a weight loss of 4.05%, followed by decompositions in the range of 95–264 °C with weight losses of 46.55% and 21.34%, due to dehydration processes and losses of volatile products, associated with endothermic peaks at 141.75 °C and 202.16 °C (Figure 15b). A next decomposition step occurs from 264 to 321 °C, due to unreacted intermediaries. The last stage of degradation above 322 °C is due to the urethanes, at a weight loss of 6.04%. This degradation also occurs due to the thermal reaction of depolymerization between carbonates and amines. In this first polymer, there is a higher rate of weight loss than polymers with nanoparticles, due to the different physical condition of the samples.

### 3.4. Characterizations of PHUs/SiO_2_ Polymeric Materials

#### 3.4.1. FTIR and UV-Vis Spectroscopies

Polymeric materials were obtained from the manufactured PHUs and different concentrations (20, 30 and 40%) of silica nanoparticles, named P2 (20%), P3 (30%) and P4 (40%), respectively. As can be seen in the polymers’ IR spectra (Figure 16), significant absorption bands at 3350 cm^−1^, 1640 cm^−1^, and 1570 cm^−1^ are representative of the urethane group [14,30] as well as the band at 1050 cm^−1^ corresponding to the Si–O–Si bonds of silica nanoparticles [22,23,24], which confirmed the successful introduction of SiO_2_ NPs in the polymeric matrix. The variation in transmittance intensity of the three polymers, but more in P3 (30%) and P4 (40%), is attributed to the presence of more functional groups in their structure, while the displacements between 1400 and 1600 cm^-1^ are due to different arrangement of the bonds of the principal chain influenced by the addition of more silica nanoparticles.

In the UV-Vis spectrum of polymers (Figure 17), the wavelength has a maximum absorption at 300 nanometers, the same as polymers with no nanoparticles, assigned to the electronic transitions of the urethane bonds [34]. Due to the fact silica nanoparticles have high absorbance by themselves (Figure 7b), the absorbance of the polymers should tend to increase with a greater concentration of nanoparticles, as occurs for P4 (40%), however, in P2 (20%) the absorbance decreases, possibly due to the material lacking homogenization with less NPs. For the purposes of this investigation, it is of interest to maintain a low energy absorbance in the material, mainly due to the fact polymers with thermal insulation applications are exposed to climatic conditions and, therefore, to the absorption of light and energy.

#### 3.4.2. Scanning Electron Microscopy

By Scanning Electron Microscope, the surfaces of polymers containing different concentrations of silica nanoparticles were analyzed. 

P2 (20%) is observed in Figure 18a,b, P3 (30%) in Figure 18c and P0 (40%) in Figure 18d. As the concentration of silica nanoparticles increases, a material of greater porosity is observed. Embedded nanoparticles were observed on the surface of all three polymers at a magnification of 1500×, due to the dispersion form in which they are found. In other magnifications, particles were seen as buried.

#### 3.4.3. Thermogravimetric Analysis and Derivate Thermogravimetric Analysis

A thermal study was performed using the TGA and DTG techniques on the fabricated polymers to know the influence of the SiO_2_ nanoparticles at different concentrations (20, 30 and 40%). An apparent simple degradation process is observed in P2 (20%); however, a multistage degradation process is appreciated in Figure 19a. In a first instance, P2 (20%) displays a weight loss of 83.44% between room temperature and 138 °C, due to the dehydration and dealcoholization processes, as to the loss of solvents, with an endothermic peak at 99.70 °C as seen in Figure 19b. The next weight loss of 10.30% up to 340 °C is assigned to the loss of non-stable amines and its respective endothermic peak at 313.9 °C and starting from 341 °C and up to 700 °C, the last decomposition of the urethane bonds occurs with an endothermic peak at 506.83 °C. In Figure 20a, P3 (30%) presents the first degradation up to 144 °C with a 61.53% of weight loss due to volatile products and its endothermic peaks can be appreciated at 97.02 and 118.55 °C in Figure 20b. In the range of 150–350 °C, the weight losses of approximately 22.13%, are due to the release of non-stable amines, urethane bonds, and carbonated chain degradation, and are assigned to endothermic peaks at 199.68 and 326.07 °C. The SiO_2_ NPs remain stable and are the last to decompose around 700 °C with a weight loss of 6.07%. For P4 (40%), a multistep degradation process also occurs, where a weight loss of 37.60% is accredited to the loss of solvents and humidity up to 222 °C (see Figure 21a) and its endothermic peaks at 41.41 and 195.07 °C. Followed by decomposition in the range of 223–531 °C of the urethane bonds with a weight loss of 12.55% and their endothermic peaks at 484.71 and 502.67 °C, as observed in Figure 21b. As well, the silica NPs are observed in the last phase with a weight loss of 3.43% from 532 °C.

The third stage of degradation of these polymers at 310 °C, occurs in a similar way to the NIPU according to the literature [36,37]. Due to the presence of hydroxyl bonds in the urethane groups of NIPU materials and lower molecular weights than traditional Pus that present their last decompositions from 450 °C [38], they are considered less thermally stable [16]. The multistage decomposition is produced by intermediates that did not react, such as amines and carbonates, and the moisture of silica nanoparticles [17] which affects the purity of the material. In general, the polymers differ a little from each other in their weight loss, which can be attributed to the fact that nanoparticles are dispersed in the polymer and not linked by a formal bond. 

Compared to P2 (20%) and P3 (30%), the endothermic effects of polymer P4 (40%) is much lower due to its higher concentration of SiO_2_ nanoparticles, in which endothermic reactions occur when the outside provides heat and then begins to degrade. Above 700 °C, no other DTG peaks are observed. As mentioned before, these polymers reinforced with SiO_2_ nanoparticles, occurs a lower rate weight loss in comparison to polymer without nanoparticles.

## 4. Conclusions

Through this investigation, detailed fabrication and analysis of the polymerization of five-membered CC with amines to form poly(hydroxy)urethanes is reported. Herein, by the proposed catalytic cyclic, the cyclic carbonates were successfully synthesized by a sol-gel method with CO_2_ addition, from an efficient process free of high energy requirements under a controlled atmosphere, with reduced reaction times and friendly to the environment. Butylene carbonate was obtained starting from epoxybutane where the reaction involves the oxidation of carbon dioxide and the reduction of LiCl. PHUs were obtained with the addition of hexamethylenediamines to both phases of cyclic carbonates in molar solution and allowing their polymerization. Times and temperatures were established according to the stoichiometric calculations proposed for the materials: (i) 5cc, (ii) PHUs, (iii) SiO_2_ nanoparticles and (iv) polymers with SiO_2_ nanoparticles, using a maximum temperature of 95 °C and a total time of 4 h. According to the different characterizations of polymers, is possible to confirm their reproducibility of the ring-opening polymerization by diamines.

The results show that polymers differ somewhat in their chemical structure from traditional NIPUs and consequently, possess improved thermal insulating capacity. We have found that the addition of SiO_2_ nanoparticles significantly improves the thermal stability of PHU materials, with homogeneous distribution and homogenous particle size distributions close to 100 nm, which after the separation process, evolved to around 68 nm. We also found that the formation of the resulting PHUs was possible due to a strong affinity between the raw materials. By this more facile and accessible manufacturing route for PHUs, in comparison with the latest advances, these films may have good potential for dull, glossy coatings or similar applications. The production of non-isocyanate polyurethane materials under mild conditions is a significant challenge for the development of more environmentally friendly materials. The manufactured polymers that were obtained in films could be scalable and useful as a dual application, for other important uses in the polyurethane industry.

## Figures and Tables

**Figure 1 polymers-11-01596-f001:**
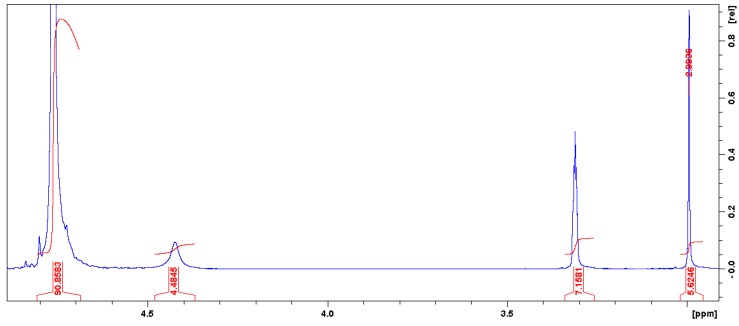
^1^H-NMR spectra of the 5cc.

**Figure 2 polymers-11-01596-f002:**
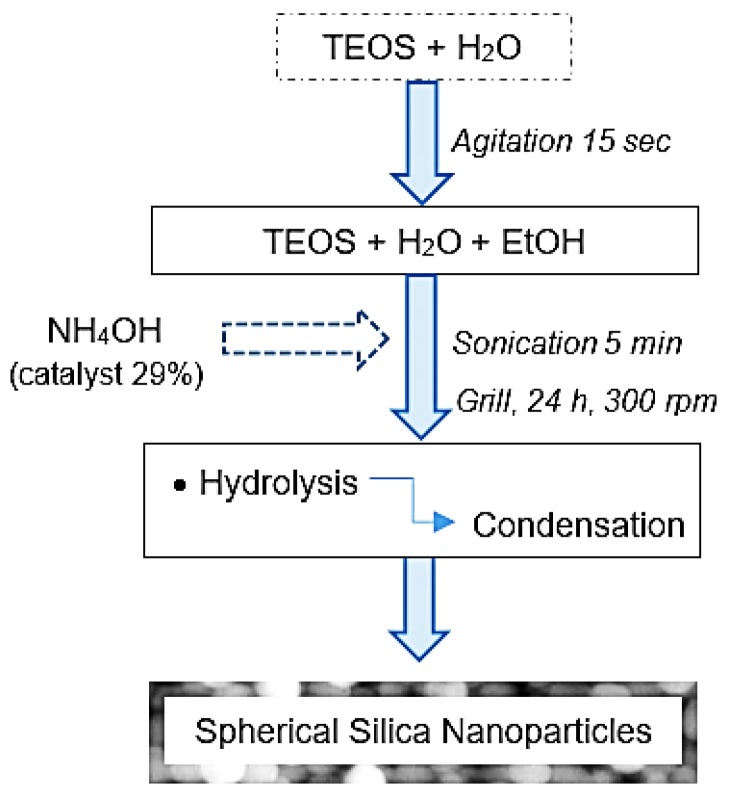
Flow chart for the synthesis of silica nanoparticles by the sol-gel method.

**Figure 3 polymers-11-01596-f003:**
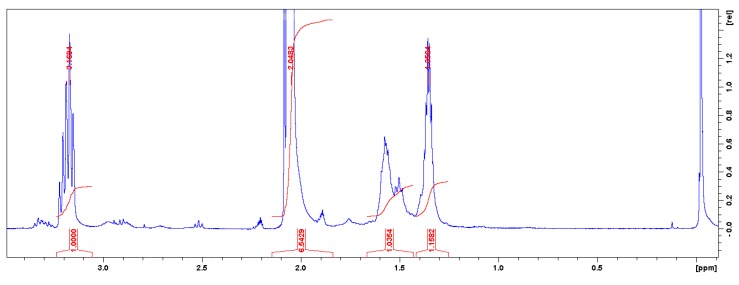
NMR spectrum of polymer -isomer a- (dissolved in CD_3_COCD_3_).

**Figure 4 polymers-11-01596-f004:**
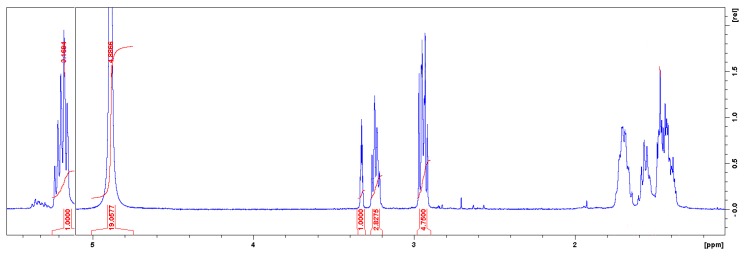
NMR spectrum of PHU-based polymer (dissolved in CD_3_OD).

**Figure 5 polymers-11-01596-f005:**
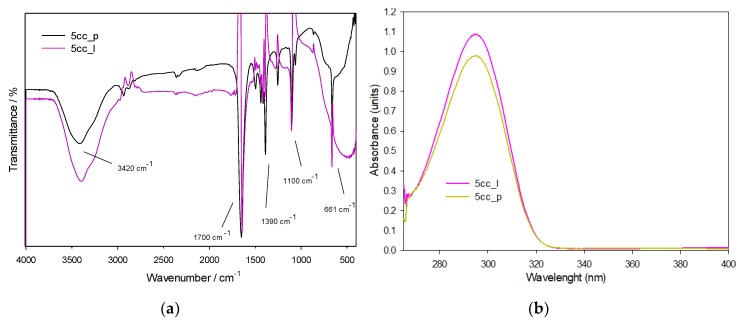
(**a**) FTIR spectrum and (**b**) UV-Vis spectrum of both phases of 5cc.

**Figure 6 polymers-11-01596-f006:**
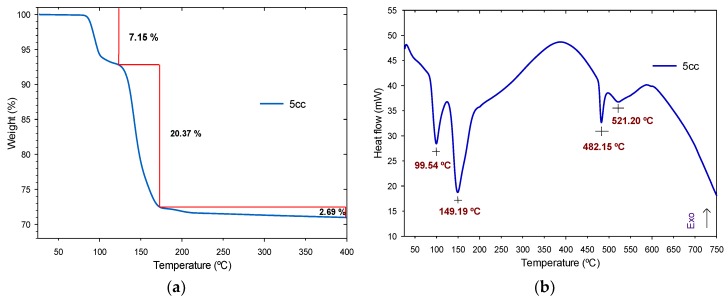
Cyclic carbonates results: (**a**) TGA curve and (**b**) DTG thermograph.

**Figure 7 polymers-11-01596-f007:**
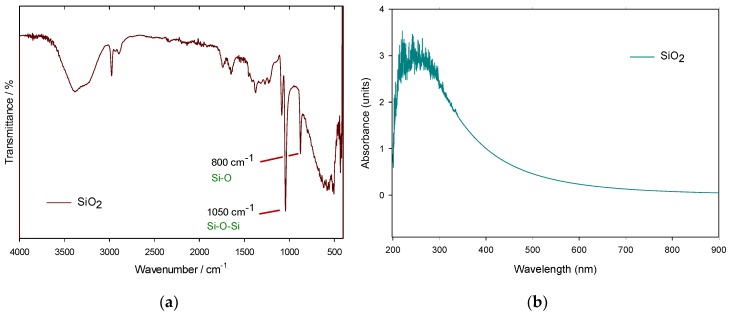
(**a**) IR spectrum and (**b**) Optical absorption spectrum of SiO_2_ NPs.

**Figure 8 polymers-11-01596-f008:**
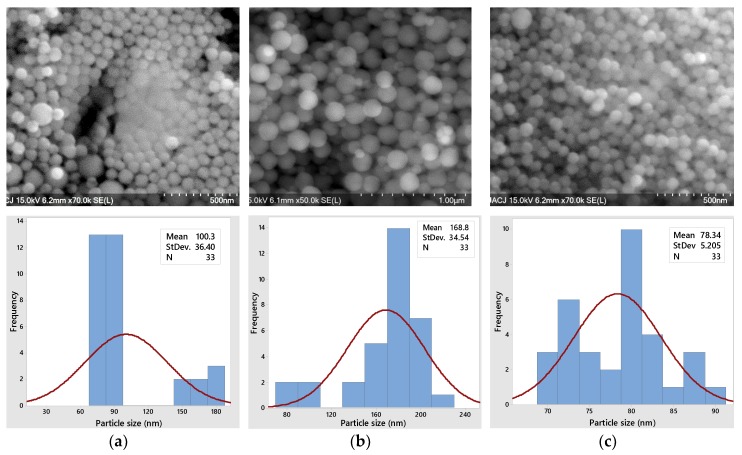
SEM images and histograms of particle size distribution of SiO_2_ NPs: (**a**) before centrifugation, (**b**) precipitated phase after centrifuge and (**c**) supernatant phase after centrifuge.

**Figure 9 polymers-11-01596-f009:**
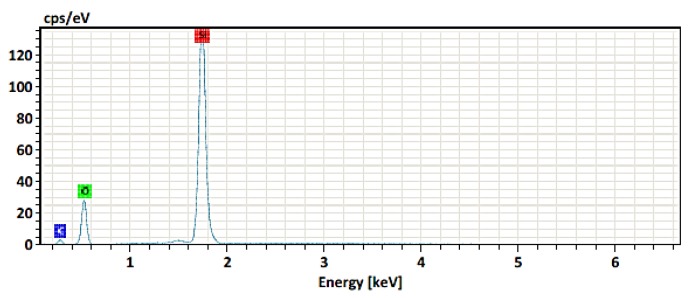
Chemical composition by EDS of the silica nanoparticles.

**Figure 10 polymers-11-01596-f010:**
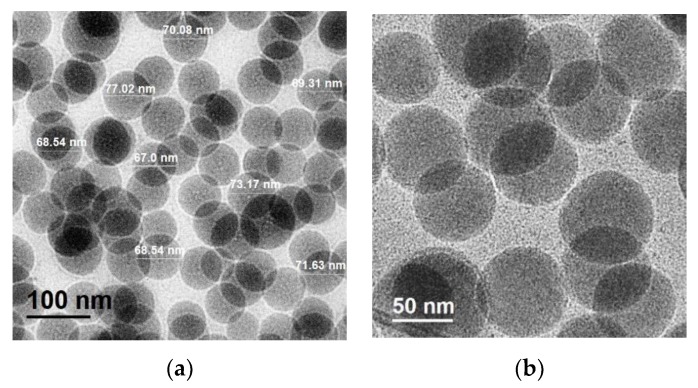
TEM images of silica NPs and their: (**a**) Particle size distribution, (**b**) spherical morphology.

**Figure 11 polymers-11-01596-f011:**
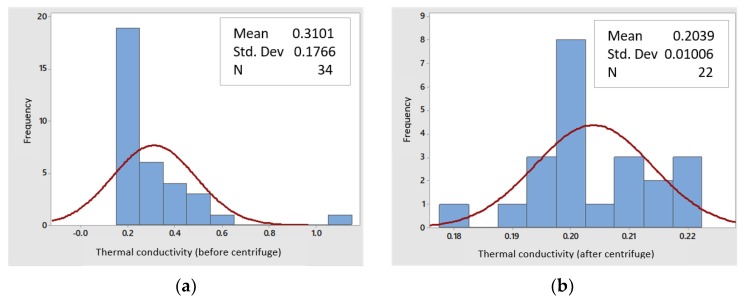
Thermal conductivity of SiO_2_ NPs (**a**) before and (**b**) after the separation process.

**Figure 12 polymers-11-01596-f012:**
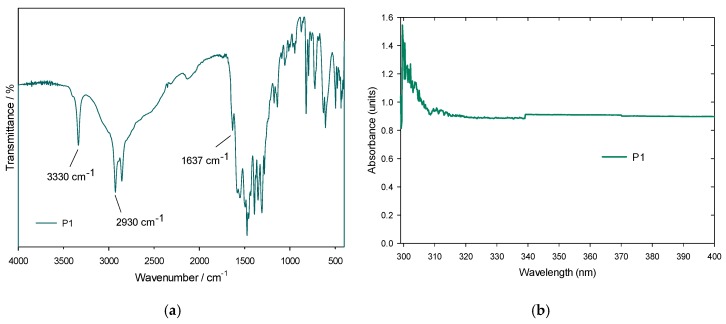
(**a**) IR spectra and (**b**) UV-Vis spectra of P1.

**Figure 13 polymers-11-01596-f013:**
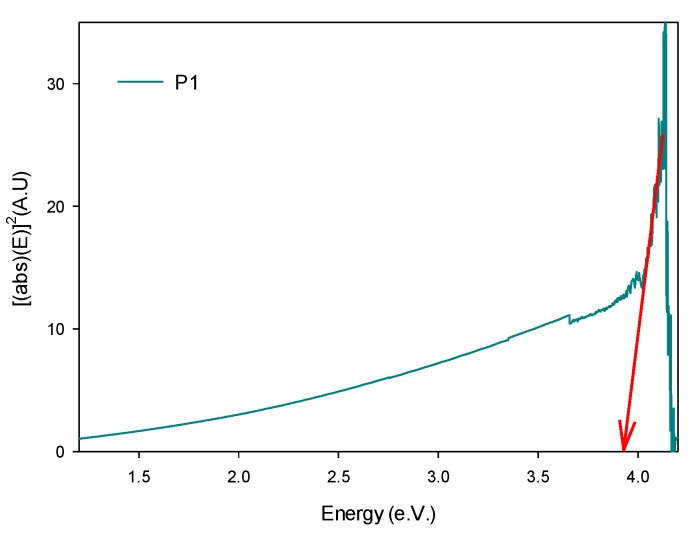
Plot of the visualization of the bandgap in PHUs.

**Figure 14 polymers-11-01596-f014:**
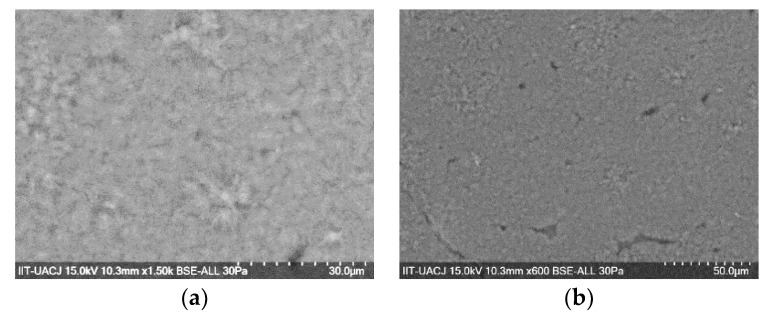
SEM images of P1 magnified to (**a**) 15,000× and (**b**) 600×, respectively.

**Figure 15 polymers-11-01596-f015:**
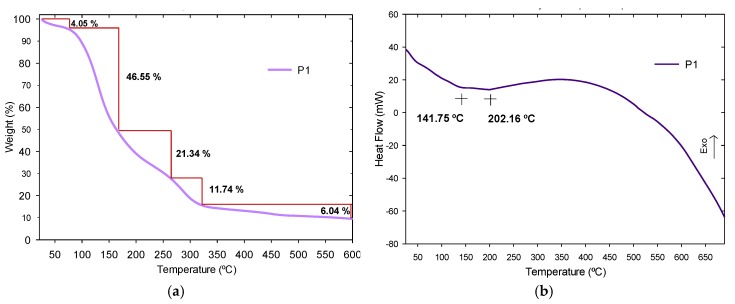
(**a**) TGA curves and (**b**) DTG curves of P1.

**Figure 16 polymers-11-01596-f016:**
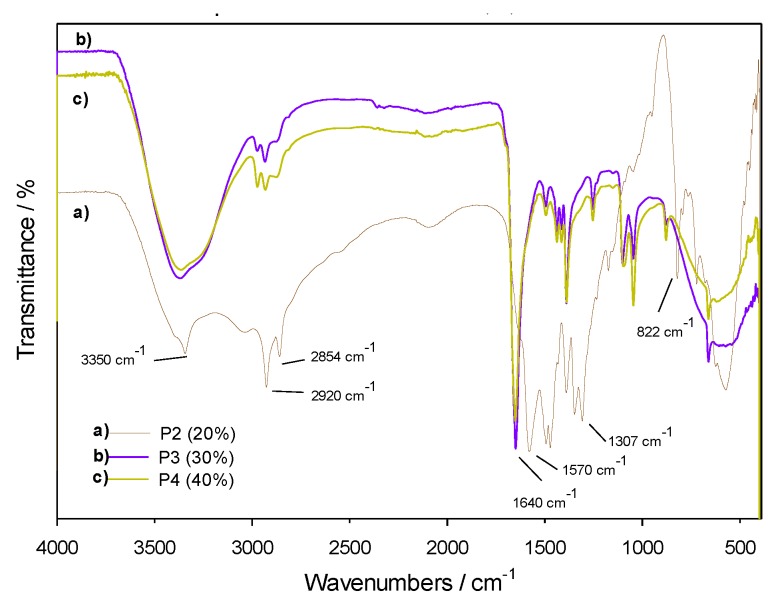
IR spectra of P2 (20%), P3 (30%) and P4 (40%).

**Figure 17 polymers-11-01596-f017:**
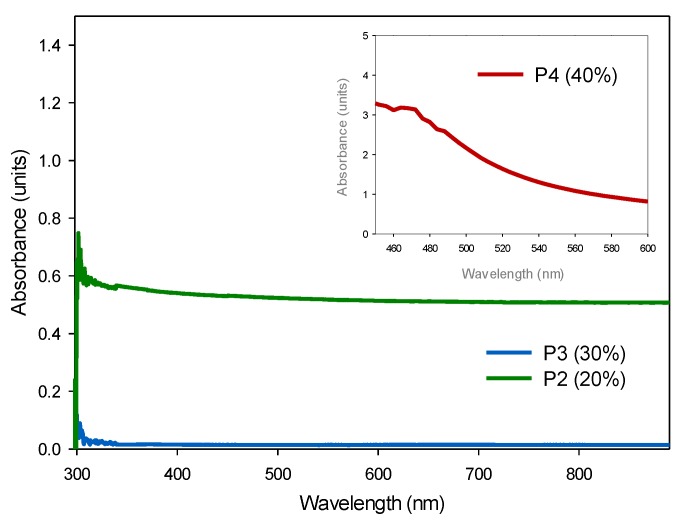
UV-Vis spectrums of P2 (20%) (Green line), P3 (30%) (Blue line) and P4 (40%) (Red line).

**Figure 18 polymers-11-01596-f018:**
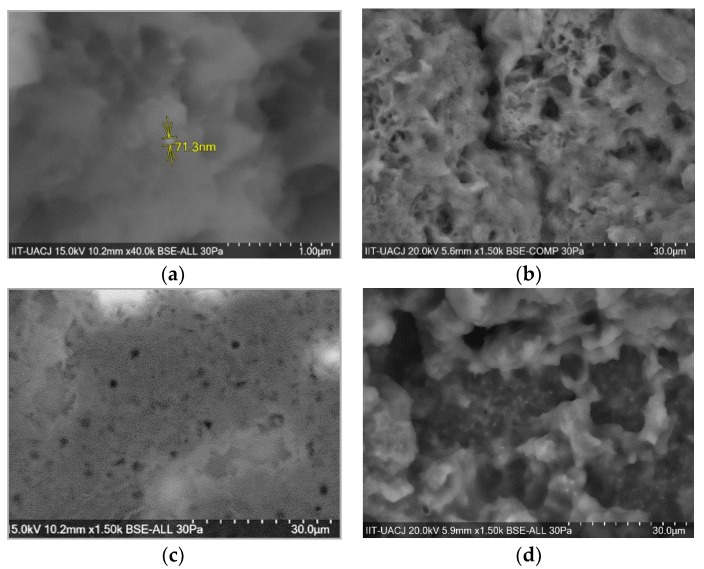
SEM images of (**a**) P2 (20%) at 40,000×, (**b**) P2 (20%) at 1500×, (**c**) P3 (30%) at 1500× and (**d**) P4 (40%) at 1500×.

**Figure 19 polymers-11-01596-f019:**
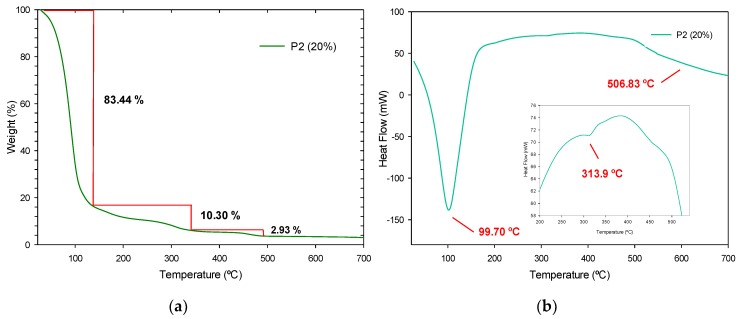
P2 (20%) results: (**a**) TGA analysis and (**b**) DTG thermogram.

**Figure 20 polymers-11-01596-f020:**
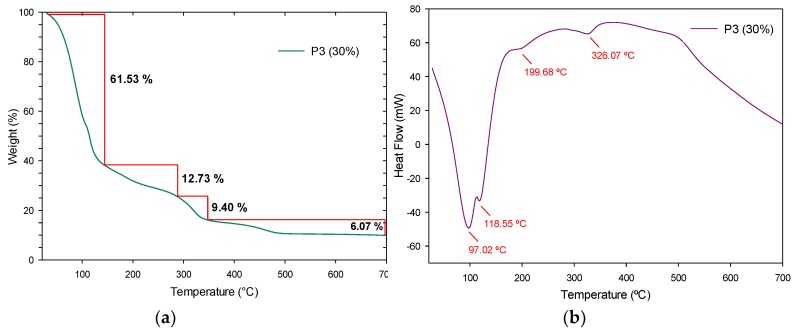
P3 (30%) results: (**a**) TGA analysis and (**b**) DTG thermogram.

**Figure 21 polymers-11-01596-f021:**
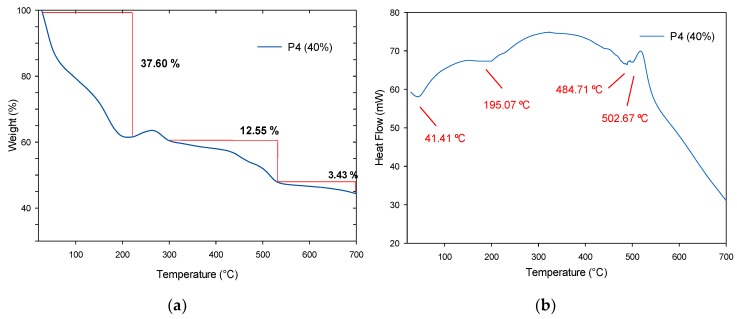
P4 (40%) results: (**a**) TGA analysis and (**b**) DTG thermogram.

**Table 1 polymers-11-01596-t001:** Material identification and their description.

Sample Identification	Sample Description
**5cc_l**	Liquid phase of cyclic carbonates
**5cc_p**	Precipitated phase of cyclic carbonates
**5cc**	Cyclic carbonates including both liquid and precipitated phase
**P1**	Poly(hydroxy)urethanes (PHUs)
**P2 (20%)**	20% of NPs concentration within the polymer (PHUs/SiO_2_)
**P3 (30%)**	30% of NPs concentration within the polymer (PHUs/SiO_2_)
**P4 (40%)**	40% of NPs concentration within the polymer (PHUs/SiO_2_)

**Table 2 polymers-11-01596-t002:** FTIR bands of cyclic carbonates as identified elsewhere [14,15,16,17,18,19].

Interaction	Bond	cm^−1^
A. Stretching	O–H	3420
B. Stretching	C–H	2950
C. Stretching	C=O	1780
D. Bending	C–H	1390
E. Stretching	O–C–O	1100
F. Stretching	O–C–O	661

**Table 3 polymers-11-01596-t003:** Elemental analysis of the SiO_2_ nanoparticles.

Element	At. No.	Line s.	Netto	Mass [%]	Mass Norm. [%]	Atom [%]	abs.error [%] (1 sigma)	abs.error [%] (2 sigma)	abs.error [%] (3 sigma)
Silicon	14	K-Serie	376313	45.63	51.88	36.10	1.92	3.84	5.76
Oxygen	8	K-Serie	53245	31.24	35.51	43.38	3.69	7.38	11.07
Carbon	6	K-Serie	5076	11.09	12.61	20.51	1.76	3.51	5.27
				**Sum 87.96**	**100.00**	**100.00**			

**Table 4 polymers-11-01596-t004:** Elemental analysis of SiO_2_ NPs.

El	AN	Series	unn. C [wt.%]	Norm. C [wt.%]	C Atom. C [at. %]	Error	(1 Sigma) [wt. %]
**O**	8	K-Series	19.52	54.61	67.87	-	3.62
**Si**	14	K-Series	16.22	45.39	32.13	-	1.01
**C**	6	K-Series	0.00	0.00	0.00	-	0.00
**Cu**	29	K-Series	0.00	0.00	0.00	-	0.00
		Total:	**35.75**	**100.00**	**100.00**		

**Table 5 polymers-11-01596-t005:** FTIR bands of PHUs as identified elsewhere [29,30,31,32,33].

Interaction	Bond	cm^−1^
A. Stretching	N–H O–H	3440
B. Stretching	C–H	2934
C. Stretching	C–H	2855
D. Stretching E. Bending	C=O N–H	1690
F. Bending	C–H	1470
G. Bending	C–H	1398
H. Stretching	C–N	1355
I. Stretching	C–O	1312

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
