# Peer review of "Production and Characterization of Non-Isocyanate Polyurethane/SiO2 Films Through a Sol-Gel Process for Thermal Insulation Applications"

_polymers, 2019, doi:10.3390/polym11101596_

Round 1

Reviewer 1 Report

Dear Authors,

Please find my comments in the enclosed file.

Author Response

Dear Reviewers:

We sincerely thank the reviewers for the time invested in reviewing our paper as well as their valuable comments. We have revised the paper taking into account the comments provided. We are now submitting the revised manuscript for review and potential publication

Reviewer 1 general comments:

The Authors discuss the synthesis of non-isocyanate polyurethanes containing silica nanoparticles for thermal insulation applications. I believe that the subject matter is interesting in terms of basic and potentially also applied research. The contributions presented in the manuscript are original and significant enough to warrant publication. That said, the manuscript contains a number of statements that I perceive as unclear or insufficiently supported by the presented experimental evidence. In my opinion, there are also some minor factors, mostly related to the presentation of the results that adversely affect the Reader’s comprehension of this interesting manuscript.

As such, I recommend the publication of the manuscript in MDPI Polymers only pending significant modification of the manuscript, with the most important issues and comments being listed below:

Response to reviewer #1

Major remarks:

Point 1) Section 2.2.1: The proposed mechanism is reasonable, however, I fail to see any experimental evidence supporting it. Is this a well-known and published mechanism for this reaction? If so, please provide the relevant references. Otherwise, please include in the manuscript (or in the supplementary information / appendix) the evidence necessary to substantiate your proposal. Should this be a mechanism that has not been confirmed beforehand and you would prefer not to include the abovementioned experimental evidence in the manuscript, please remove the entirety of this section.

Author response: Done. Authors prefer remove the section

Minor remarks:

Point 2) Lines 42-46: The transition between your criticism of the traditional polyurethanes, due to the high embedded energy in their manufacturing, and the conclusion that the non-isocyanate polyurethanes (NIPUs) are in demand is rather abrupt. I suggest providing a bit more background on NIPUs; maybe a comparison of the energies used in the two processes would be in order to better illustrate your point to the Reader?

Author response: Authors employed the sol gel method to produce NIPU, that process directly impacts on time, energy and costs mainly due to isn’t necessary to used reactors, long time of process an vacumm or higer temperatures

Point 3) Lines 78-80: For dimethylformamide, you are giving its purity grade twice - in parentheses and along with the purity grades for other chemicals.

Author response: Done

Point 4) Line 80: Please correct the formula for poly(vinyl alcohol) - I suggest either (C2H3O)n or (CH2CHOH)n

Author response: We used (CH2CHOH)n

Point 5) Figure 1: The sizes of the oxygen atoms in several instances of the PVAl structure are not identical.

Author response: Figure 1 not contain oxygen atoms, the figure is RMN spectra of the 5cc and the mechanism with oxygen was removed

Point 6) Figure 5b: A typographical error in the x axis label - it should be "Wavelength"

Author response: Figure 6(b) is a UV-Vis spectrum of 5cc with X axis as Wavelength

Point 7) The English used in the manuscript, apart from some minor mistakes, is generally satisfactory.

Author response: Finally, about English language and style: We have reviewed the manuscript carefully and made changes were considered necessary. We believe the improved grammar and punctuation will better serve the reader.

Reviewer 2 Report

Dear Editor,

Recommendation: Major revisions needed as noted before publishing the manuscript.

The manuscript “Production and characterization of non-isocyanate polyurethane/SiO2 films through a sol-gel process for thermal insulation applications” by Dr. H. Mota and his co-workers reports a composite material of polyurethane made from safe raw materials and silicon dioxide by sol-gel method. I think this paper is valuable because so many experimental results have been shown and thermal management is an area of great need in various field. However, on the other hand, this paper seems to have some big problems. I think that it is necessary to improve the following points.

If this paper is published on Polymers, a major revision is required.

Major issues,

It is not clear what Authors want to insist on in this paper regarding the introduction, experimental and results and discussion. It looks like an evaluation report rather than a scientific paper.

In addition, many figures and tables are created by copy and paste from the measuring device. For example, in FIGS. 4 and 5, unnecessary information is described in the upper left.

Minor issues,

1, L71-74 should not delete from introduction.

2, L93; LiCl solution (“1M”) should change to the unit of concentration.

3, L80; “C2H3OR” is not the chemical formula of poly(vinyl alcohol).

4, L184; The notation “P1” for pure PHUs is confusing. Should it be P0 at least?

5, P338; Figure caption of Figure 11 is not clear.

6, L409; Can DTA measure heat flow? Normally, to measure heat flow, DSC is required.

Author Response

Dear Reviewers:

We sincerely thank the reviewers for the time invested in reviewing our paper as well as their valuable comments. We have revised the paper taking into account the comments provided. We are now submitting the revised manuscript for review and potential publication

Reviewer 2 general comments:

The manuscript “Production and characterization of non-isocyanate polyurethane/SiO2 films through a sol-gel process for thermal insulation applications” by Dr. H. Mota and his co-workers reports a composite material of polyurethane made from safe raw materials and silicon dioxide by sol-gel method. I think this paper is valuable because so many experimental results have been shown and thermal management is an area of great need in various field. However, on the other hand, this paper seems to have some big problems. I think that it is necessary to improve the following points.

If this paper is published on Polymers, a major revision is required.

Response to reviewer #2

Major remarks:

Point 1) It is not clear what Authors want to insist on in this paper regarding the introduction, experimental and results and discussion. It looks like an evaluation report rather than a scientific paper.

In addition, many figures and tables are created by copy and paste from the measuring device. For example, in FIGS. 4 and 5, unnecessary information is described in the upper left.

Author response: Done. Authors have attended the reviewer recommendation

Minor remarks:

Point 2) L71-74 should not delete from introduction.

Author response: We changed the paragraph of the introduction under materials.

Point 3) L93; LiCl solution (“1M”) should change to the unit of concentration.

Author response: The solution remained under constant stirring for an additional twenty minutes at a temperature of 95 °C in the presence of LiCl as a catalyst and ending after another 30 minutes.

Point 4) L80; “C2H3OR” is not the chemical formula of poly(vinyl alcohol).

Author response: We changed as (CH2CHOH)n as reviewer proposed.

Point 5) L184; The notation “P1” for pure PHUs is confusing. Should it be P0 at least?

Author response: Authors identified the first polymer as P1 without nanoparticles and P2 (20%), P3 (23%) y P4 (40%) whit different concentrations of SiO2 nanoparticles, due to P2 to P4 are consistent with the concentration employed.

Point 6) P338; Figure caption of Figure 11 is not clear.

Author response: Figure 10. a) Particle size distribution, b) spherical morphology.

Point 7) L409; Can DTA measure heat flow? Normally, to measure heat flow, DSC is required.

Author response: We changes DTA AS DTG in Figure 17. (a) TGA curves and (b) DTA curves of P1.

Point 8) English language and style are fine/minor spell check required

Author response: Finally, about English language and style: We have reviewed the manuscript carefully and made changes were considered necessary. We believe the improved grammar and punctuation will better serve the reader.

Round 2

Reviewer 1 Report

Dear Authors,

You have done good work and addressed all my remarks - I have no further comments and recommend your manuscript for publication in MDPI Polymers.

Reviewer 2 Report

The authors are responding appropriately to comments, and I think that it is safe to publish the current manuscript.